# Immune Functions of Signaling Lymphocytic Activation Molecule Family Molecules in Multiple Myeloma

**DOI:** 10.3390/cancers13020279

**Published:** 2021-01-13

**Authors:** Mariko Ishibashi, Rimpei Morita, Hideto Tamura

**Affiliations:** 1Department of Microbiology and Immunology, Nippon Medical School, Tokyo 113-8602, Japan; rimpei-morita@nms.ac.jp; 2Division of Diabetes, Endocrinology and Hematology, Department of Internal Medicine, Dokkyo Medical University Saitama Medical Center, Saitama 343-8555, Japan; tam@nms.ac.jp; 3Department of Hematology, Nippon Medical School, Tokyo 113-8602, Japan

**Keywords:** multiple myeloma, immunotherapy, SLAM family, SLAMF2 (CD48), SLAMF3 (CD229, Ly9), SLAMF6 (CD352, NTB-A), SLAMF7 (CD319, CRACC, CS1)

## Abstract

**Simple Summary:**

Multiple myeloma (MM) is an incurable hematological malignancy characterized by an increase in abnormal plasma cells. Disease progression, drug resistance, and immunosuppression in MM are associated with immune-related molecules, such as immune checkpoint and co-stimulatory molecules, present in the tumor microenvironment. Novel agents targeting these cell-surface molecules are currently under development, including monoclonal antibodies, bispecific monoclonal antibodies, and chimera antigen receptor T-cell therapies. In this review, we focus on the signaling lymphocytic activation molecule family receptors and provide an overview of their biological functions and novel therapies in MM.

**Abstract:**

The signaling lymphocytic activation molecule (SLAM) family receptors are expressed on various immune cells and malignant plasma cells in multiple myeloma (MM) patients. In immune cells, most SLAM family molecules bind to themselves to transmit co-stimulatory signals through the recruiting adaptor proteins SLAM-associated protein (SAP) or Ewing’s sarcoma-associated transcript 2 (EAT-2), which target immunoreceptor tyrosine-based switch motifs in the cytoplasmic regions of the receptors. Notably, SLAMF2, SLAMF3, SLAMF6, and SLAMF7 are strongly and constitutively expressed on MM cells that do not express the adaptor proteins SAP and EAT-2. This review summarizes recent studies on the expression and biological functions of SLAM family receptors during the malignant progression of MM and the resulting preclinical and clinical research involving four SLAM family receptors. A better understanding of the relationship between SLAM family receptors and MM disease progression may lead to the development of novel immunotherapies for relapse prevention.

## 1. Introduction

Multiple myeloma (MM) is a B-cell malignancy that arises from the clonal expansion of aberrant plasma cells (MM cells). In addition to the accumulation of clonal plasma cells, MM is characterized by elevated levels of monoclonal protein (M-protein), renal failure, hypercalcemia, anemia, and bone lesions [1,2]. This malignancy is caused by multiple genetic alterations and microenvironmental changes in bone marrow (BM) cells and nearly always develops from monoclonal gammopathy of undetermined significance (MGUS), a precancerous condition [1,3,4,5].

In the past several years, the treatment options for patients with refractory MM have advanced dramatically due to the emergence and approval of pharmaceuticals such as immunomodulatory drugs (IMiDs; e.g., lenalidomide and pomalidomide), proteasome inhibitors (e.g., bortezomib, carfilzomib, and ixazomib), and monoclonal antibodies (e.g., elotuzumab, daratumumab, and isatuximab) [1,6,7,8,9,10]. Many clinical trials of immunotherapies for MM are ongoing, with a particular focus on IMiDs, monoclonal antibodies, bispecific antibodies, immune checkpoint inhibitors, and chimeric antigen receptor (CAR) T-cell therapy. Several molecules have been selected as candidate targets for MM immunotherapies, including signaling lymphocytic activation molecule (SLAM) family 7 (SLAMF7) and CD38, which are targeted by elotuzumab and daratumumab/isatuximab, respectively, and other cell-surface antigens such as B-cell maturation antigen (BCMA), CD56, CD138, CD74, interleukin (IL)-6 receptor, vascular endothelial growth factor (VEGF), and activated conformation of integrin β7 [11,12,13].

Promising therapeutic target antigens for MM must have the following characteristics: (1) specific or strong expression on the surfaces of MM cells relative to normal immune cells and tissues; (2) expression on MM progenitor cells or cancer stem cells but not on normal hematopoietic stem cells (HSCs) and hematopoietic progenitor cells (HPCs); (3) constitutive strong expression, regardless of disease progression or treatment; and (4) an association with the biological functions or molecular pathogenesis of MM (e.g., cell survival and growth, antiapoptotic responses, angiogenesis, cell–cell communications, and interactions with BM stromal cells). Therefore, an understanding of the expression and biological functions of these molecules in MM is important to support the development of novel therapeutic targets and the prevention of treatment-related side effects.

## 2. Overview of SLAM Family Receptors

SLAM family receptors play important roles in the immune responses mediated by a broad range of innate and adaptive immune cells [14,15,16,17,18]. The SLAM family comprises nine members: SLAMF1 (CD150); SLAMF2 (CD48); SLAMF3 (CD229, Ly9); SLAMF4 (CD244, 2B4); SLAMF5 (CD84); SLAMF6 (CD352, NTB-A, SF2000); SLAMF7 (CD319, CS1, CRACC); SLAMF8 (CD353); and SLAMF9 (CD84H1, SF2001) (Figure 1). All except for SLAMF2 are type I transmembrane glycoproteins within the immunoglobulin (Ig) superfamily and are composed of extracellular IgV- and IgC2-like domains with high self-affinity through the IgV-like domain. In contrast, SLAMF2 interacts with SLAMF4 [14,19,20]. The cytoplasmic tail of a SLAM family receptor contains an immunoreceptor tyrosine-based switch motif (ITSM), TxYxxI/V, where T, Y, I, V, and x represent threonine, tyrosine, isoleucine, valine, and any other amino acid, respectively (Figure 1) [19,20]. Self-ligated SLAM family receptors recruit Src homology region 2 (SH2) domain-containing adaptor proteins such as SLAM-associated protein (SAP; SH2D1A) or Ewing’s sarcoma-associated transcript 2 (EAT-2; SH2D1B) to the phosphorylated tyrosine within the ITSM sequence. SAP is expressed in T cells, natural killer (NK) cells, NKT cells, eosinophils, and platelets but is absent or minimally expressed in some B cell populations, macrophages, and dendritic cells (DCs) [16,21]. EAT-2 is expressed in NK cells and antigen-presenting cells (APCs) including B cells, macrophages, and DCs [22,23]. In SLAM family receptors containing ITSMs in the cytoplasmic domain (SLAMF1 and SLAMF3–SLAMF7), SLAM–SAP, and SLAM–EAT-2 complexes interact directly with Src family kinase Fyn and phospholipase Cγ (PLCγ), respectively, and trigger an activating signal within these immune cells (Figure 1). However, the signal transduction pathway mediated by SLAMF2, SLAMF8, and SLAMF9, which lack intercellular signaling motifs, is still unknown. Further, SAP and EAT-2 inhibit the recruitment of inhibitory adaptor proteins such as SH2 domain phosphatase-1 (SHP-1), SHP-2, SH-2 domain inositol 5ʹ-polyphosphatase 1 (SHIP1), or Csk to ITSMs of SLAM family receptors and, in the absence of SAP and EAT-2, the self-ligated receptor transduces an inhibitory signal via these inhibitory adaptor proteins [24,25,26].

SLAM family molecules are expressed in various hematological malignancies (e.g., MM, chronic lymphocytic leukemia [CML], acute myeloid leukemia [AML], lymphoma) and solid tumors (e.g., hepatocellular carcinoma, central nervous system tumors, renal cancer) [19]. In MM patients, malignant plasma cells strongly express SLAMF2, SLAMF3, SLAMF6, and SLAMF7. In contrast, SLAMF1, SLAMF4, and SLAMF5 expression is limited and SLAMF8 and SLAMF9 have not been reported in this malignancy [27,28,29,30,31]. The genes encoding SLAM family receptors and EAT-2 are clustered in the long arm of chromosome 1 (1q23) [17,32]. In contrast, the gene encoding SAP is localized on the X chromosome (Xq25), and X-linked lymphoproliferative disease (XLP), a rare heritable immunodeficiency disorder, is associated with SAP deficiency [32,33]. Although 36% of newly diagnosed MM patients present with a gain or amplification of the long arm of chromosome 1 (+1q; region, 1q21–1q23.1) [34,35,36], the relationship between +1q and the level of SLAM family receptor expression has not been explored previously. This review focuses on the expression and biological functions of SLAMF2, SLAMF3, SLAMF6, and SLAMF7 in both normal immune cells and MM cells and discusses their potential usefulness as therapeutic targets in relapsed/refractory MM.

## 3. SLAMF2 (CD48)

SLAMF2 is a glycosylphosphatidylinositol (GPI)-anchored glycan of the CD2 superfamily of Ig-like receptors and is expressed on nearly all hematopoietic cells [37]. Unlike other SLAM family molecules, SLAMF2 facilitates cell–cell communication by binding to its ligands CD2 and SLAMF4 (also known as CD244 and 2B4) on other immune cells, such as T, B, and NK cells [37,38,39]. The binding affinity of human SLAMF2 for SLAMF4 is 10-fold higher than its affinity for CD2 [39]. In T cells, SLAMF2 binds to GPI in cell membrane lipid rafts, where it activates leukocyte *C*-terminal Src kinase (Lck) to promote T cell receptor (TCR) signaling and activation (Figure 2A) [40]. When expressed on APCs such as DCs, SLAMF2 serves as a strong co-stimulatory molecule to enhance the cytotoxicity of SLAMF4-expressing effector/memory CD8^+^ T cells, and this cell–cell interaction prolongs DC survival [37,41]. Furthermore, SLAMF2 expressed on DCs activates SLAMF4-expressing NK cells via PLCγ activation, Ca^2+^ flux induction, and SAP- or EAT-2-mediated ERK activation.

In the absence of SAP and EAT-2, the SLAMF2–SLAMF4 interaction induces an inhibitory signal in NK cells via SHP1, SHP2, and SHIP-1 [17,26,42,43,44]. SLAMF2 and SLAMF4 on immune cells are upregulated under inflammatory conditions of the tumor microenvironment and chronic viral infection [37,44]. SLAMF4-overexpressing T cells exhibit a phenotype of exhausted CD8^+^ T cells which expresses immune checkpoint inhibitory receptors such as PD-1, LAG-3, CD160, CTLA-4, and TIM-3 and decreases IL-2 and interferon (IFN)-γ production [44]. Inflammatory conditions can induce an imbalance in the expression of SLAMF4 and SAP (e.g., high SLAMF4 and normal SAP) which suppresses T cell activation and induces T cell exhaustion (Figure 2B). Exhausted T and NK cells have been observed in various malignancies, including MM, melanoma, hepatocellular carcinoma, and acute myelocytic leukemia (AML) [45,46,47,48]. Zelle-Rieser et al. showed that SLAMF4-positive exhausted T cells exhibited characteristics such as a lower proliferative capacity and impaired function in the MM microenvironment [45]. Therefore, the SLAMF2–SLAMF4 interaction has both co-stimulatory and co-inhibitory functions in immune cells, depending on the status of SAP or EAT-2 expression (Figure 2B).

Hosen et al. reported that SLAMF2 was strongly expressed on cancer cells from 22 of 24 patients with MM but not on normal HSCs and HPCs from those individuals [49]. In that study, an antihuman CD48 antibody exerted cytotoxic effects against MM cells via NK cell-mediated antibody-dependent cell-mediated cytotoxicity (ADCC) and complement-dependent cytotoxicity (CDC) but did not target CD34^+^ HSCs/HPCs [49]. The antibody eliminated human myeloma cell lines implanted under the skin of immunodeficient mice [49]. Furthermore, in a study of 79 patients with MM, Ashour et al. observed the maintenance of high SLAMF2 expression levels when comparing cases in the following categories: newly diagnosed versus relapsed/refractory MM; Revised International Staging System (R-ISS) stage I/II versus stage III; with versus without intensive treatment; and responsiveness versus nonresponsiveness to bortezomib or IMiDs [50]. Given this broad expression of SLAMF2 in most MM patients, Lewis and colleagues evaluated the cytotoxic effect of SGN-CD48A, a CD48-targeting antibody–drug conjugate (ADC) with a glucuronide–monomethylauristatin E (MMAE) linker [51]. Exposure to SGN-CD48A potently induced G2/M cell cycle arrest in MM cells in vitro and led to complete remission in nearly all tested U266 or NCI-H929 xenograft-bearing mice but had minimal effects on human B, T, and NK cells [51]. Interestingly, however, SGN-CD48A killed both CD48^+^ and CD48^KO^ MM cells in an admixed co-culture, suggesting a potential cytotoxic bystander effect mediated by the activation of immune cell death [52]. The combination of SGN-CD48A and daratumumab yielded increased antitumor activity relative to SGN-CD48 alone or in combination with elotuzumab in MM xenograft mouse models [52]. Those results suggest that SGN-CD48A may exert cytotoxicity against MM cells through multiple direct and indirect mechanisms and may be a candidate for combination therapy with daratumumab. A clinical study of SGN–CD48A monotherapy for relapsed/refractory MM patients is in progress after patient recruitment ended (NCT02954796). Although these SLAMF2-targeting therapies induced significant effects in preclinical studies, it will be necessary to clarify whether the expression of SLAMF2 on MM cells is associated with the pathogenesis of this malignancy.

## 4. SLAMF3 (CD229, Ly9)

SLAMF3 is expressed strongly on most MM cell lines, and Atanackovic et al. identified this receptor as the most intensely expressed phosphorylated immunoreceptor in a human phosphoimmunoreceptor array involving the MM cell line MOLP-8 [27]. SLAMF3 is also expressed on Sca-1^+^c-kit^+^Lin^−^ HSCs and mainly localizes to T and B lymphocytes and NK and NKT cells (Figure 1) [53,54,55]. In contrast to the other SLAM family receptors, SLAMF3 contains a unique extracellular domain that comprises two tandem repeats of IgV–IgC2 and interacts homologically with itself via the *N*-terminal IgV domain (Figure 3A) [54,56]. The cytoplasmic domain of SLAMF3 contains two ITSMs. In T cells, SLAMF3 recruits SAP through its SH2 domain, and SAP then interacts with the Src family kinase Fyn to regulate protein kinase Cθ (PKCθ)–NF-κB pathway activity (Figure 3A) [55,57,58,59]. In the presence of an anti-mouse SLAMF3 antibody (Ly9.ab3), the inability of SLAMF3 to self-ligate suppresses the expression of the activation markers CD25 and CD69 on anti-CD3/anti-CD28 antibody-stimulated murine T cells and further decreases the secretion of IFN-γ, IL-2, IL-4, IL-6, IL-10, and tumor necrosis factor (TNF)-α [53]. SLAMF3 is associated with enhanced T cell activation and helper T cell type 2 (Th2) polarization, and consequently SLAMF3 deficiency markedly reduces T cell proliferation and IL-2 production and causes a mild Th2 defect [60]. Thus SLAMF3 is associated with enhanced T cell activation and promotes Th2 polarization.

SLAMF3 is expressed at higher levels on malignant plasma cells from MM patients than on other BM hematopoietic cells in the same samples [27,61,62,63]. However, SLAMF3 is also strongly expressed on normal plasma cells from MM patients and healthy controls [27,61,62,63,64]. Yousef et al. demonstrated that SLAMF3 expression levels were unaltered in 17 patients with MGUS, 49 with MM, 7 with smoldering MM (SMM), and 4 with plasma cell leukemia [62]. Similarly, we observed that the expression of SLAMF3 remained constitutively strong, regardless of disease stage, in a study of 17 patients with MGUS, 153 with newly diagnosed MM (19 asymptomatic and 134 symptomatic), and 30 with relapsed/refractory MM. In contrast, there were significant decreases in the expression of the plasma cell-surface antigen CD138 with disease progression [63]. Atanackovic et al. used small interfering RNA (siRNA) to demonstrate that SLAMF3 knockdown reduced the proliferation and survival of MM cell lines and enhanced apoptosis induced by anti-MM agents [27]. Similarly, we confirmed that SLAMF3 knockdown and knockout, which were, respectively, achieved using small hairpin RNA (shRNA) and the CRISPR/Cas9 system, nearly abolished the aggressive phenotypic characteristics of MM, while SLAMF3 overexpression promoted those characteristics [63].

Notably, MM cell lines do not express SAP and EAT-2, and therefore SLAMF3 signaling pathway activity is closely related to the recruitment of the adaptor proteins SHP2 and growth factor receptor-bound protein 2 (GRB2), which are expressed in most cell lines [63]. SHP2 and GRB2, respectively, bind to the two ITSMs and a *C*-terminal end motif (amino acid sequence: YENE) in the cytoplasmic domain of SLAMF3 and interact with each other (Figure 3B) [63]. When expressed on MM cells, SLAMF3 transmits MAPK/ERK signals mediated by the recruitment of SHP2 and GRB2 via self-ligand interactions and subsequently promotes antiapoptotic responses and cell proliferation (Figure 3B) [63]. The GRB2-mediated signaling cascade promotes cell proliferation and survival in multiple types of malignancy [65]. GRB2 is expressed much more strongly in plasma cells from MM patients relative to healthy controls, despite nearly equal SLAMF3 expression levels [63]. These results suggest that the SLAMF3–SHP2–GRB2 signaling pathway strongly promotes the aggressive MM cell phenotype.

The gene encoding SLAMF3 contains a single nucleotide polymorphism (SNP), rs509749 (1804A>G, M602V), which causes a nonsynonymous change from M (codon ATG) to V (codon CTG) at amino acid residue 602 (M602V) within ITSM1 of the intracellular domain. The frequencies of the rs509749 A and G alleles are 0.34 and 0.66, respectively, in the global population. The G allele is the major type in all but European populations, in which the frequencies are nearly equal (A = 0.55, G = 0.45) [66]. An association between the A allele and susceptibility to systemic lupus erythematosus (SLE) has been reported in British and Canadian populations [67,68]. We studied 68 Japanese patients with MM to determine the rs509749 allelic frequency in this population. The G allele was dominant (G = 0.76), and the frequencies of the GG, GA, and AA genotypes were 61.8%, 29.4%, and 8.8%, respectively, which were very similar to those in healthy Japanese controls [69]. Interestingly, MM patients harboring the rs509749 GG genotype had significantly shorter overall survival durations than those harboring the GA and AA genotypes (*p* = 0.046). We further determined that in MM cells, the A allele of rs509749 was associated with a markedly lower SLAMF3 binding affinity for SHP2 and GRB2 when compared with the G allele, and this reduced affinity led to decreases in ERK activation and aggressive MM behaviors [69]. Therefore, the G allele of the SLAMF3 SNP rs509749 might be a risk factor for MM development but not for plasma cell tumorigenesis.

We further investigated whether SLAMF3 may be a promising serum biomarker in MM patients. Notably, we observed elevated serum concentrations of soluble SLAMF3 (sSLAMF3) with MM progression, which led to markedly high levels in patients with advanced-stage disease [63]. sSLAMF3 is produced by cleavage of the extracellular domain from SLAMF3-expressing MM cells via matrix metalloproteinase-9 (MMP-9) or other enzymatic mechanisms [63]. MM patients harboring high circulating levels of sSLAMF3 exhibit more aggressive clinical characteristics and a significantly shorter progression-free survival (PFS) duration than those with low levels. These results suggest that the serum level of sSLAMF3 might reflect the progression of MM and could thus be a useful new prognostic marker.

SLAMF3 is expressed on MM cells regardless of disease progression or cell phenotype and is detected not only on CD38^+^CD138^+^ plasma cells but also on aberrant CD56^+^ plasma cells and CD38^+^CD138^low/negative^ chemotherapy-resistant myeloma progenitor cells [62,63]. Therefore, SLAMF3 might be useful as both a cell-surface marker and an immunotherapeutic target in MM. A monoclonal anti-SLAMF3 antibody was shown to induce MM cell lysis effectively via ADCC and CDC [27]. Moreover, the blockade of SLAMF3-mediated self-ligation between MM cells suppresses cell proliferation and facilitates melphalan-induced apoptosis [63]. Atanackovic’s group developed an innovative CAR-T cell therapy specific for SLAMF3 on MM cells [70]. Those CAR-T cells efficiently killed SLAMF3^high^ plasma cells but not SLAMF3^low^ T cells or native B cells. Interestingly, SLAMF3 CAR-T cells more effectively eliminated SLAMF3-positive, BCMA-negative memory B cells (which include putative clonotypic MM progenitor cells) when compared with BCMA CAR-T cells [70]. Thus, SLAMF3-targeted CAR-T therapy could potentially prevent posttreatment MM relapse, unlike BCMA CAR-T therapy. SLAMF3-targeted strategies have not yet proceeded to preclinical trials, and the potential for strong and promising effects in MM patients remains to be demonstrated.

## 5. SLAMF6 (CD352, NTB-A)

SLAMF6 is mainly expressed on B, T, and NK cells and positively regulates the former two populations via hemophilic receptor signaling mediated by phosphorylated SAP [17,42,71]. SLAMF6 augments T cell activation via its interaction with the TCR and subsequent phosphorylation of ERK [72]. SLAMF6 is upregulated on activated T cells, and co-stimulation via this receptor promotes the proliferation of Th1 cells and the production of cytokines such as IFN-γ and TNF-α [73,74]. CD4^+^ T cells regulate B-cell survival via SLAMF6–SLAMF6 interactions, which modulate the interactions between T follicular helper cells and germinal B cells [75]. In previous studies, SLAMF6 expression was detected on the MM cells of 13 of 15 (87%) tested patients and on some MM cell lines by flow cytometry [27,73,76]. Lewis et al. evaluated the therapeutic effects of SGN-CD352A, a humanized anti-SLAMF6 antibody (h20F3ec) conjugated with a pyrrolobenzodiazepine (PBD) dimer, against MM [76]. Notably, SGN-CD352A exerts cytotoxic activity against MM cells via the incorporation of highly cytotoxic PBD into the DNA but has minimal adverse effects on normal T and B cells. Experimentally, the administration of SGN-CD352A, but not the unconjugated h20F3ec antibody, led to complete remission in nearly all tested U266 or MM.1R xenograft-bearing mice [76]. A clinical study to determine the safety of SGN-CD352A in patients with relapsed/refractory MM is ongoing (NCT02954796). The expression patterns of SLAMF6 across various stages of MM remain unknown and must be resolved before SLAMF6 targeted therapy can be applied in clinical practice settings.

## 6. SLAMF7 (CD319, CS1)

SLAMF7 is strongly expressed by NK, NKT, and T cells and, like SLAMF4, exerts both activating and inhibitory effects depending on the expression status of SAP or EAT-2 [77]. In NK cells, SLAMF7 recruits EAT-2 but not SAP to phosphorylated tyrosine 281 (Y281) within its ITSM via homotypic interactions. This recruitment activates the PLCγ and PI3K signaling pathway to enhance NK-cell activation and cytotoxicity [23,78,79].

Hsi and colleagues identified the strong constitutive expression of SLAMF7 on CD138^+^ plasma cells from 24 healthy controls and 14, 35, and 532 patients with MGUS, SMM, and MM, respectively, through comprehensive microarray analysis [29]. Strong SLAMF7 expression was maintained regardless of disease progression, recurrence, and cytogenetic abnormalities [29,50]. The expression level of SLAMF7 on MM cells from patients was found to be >3-fold higher than that on NK cells but was equivalent to the expression level on normal plasma cells from healthy controls [29].

Treatment with a SLAMF7-targeted siRNA or anti-SLAMF7 antibody (HuLuc63) was shown to inhibit the adhesion of MM cells to BM stromal cells, which reduced the survival of MM cells (Figure 4A) [80]. Further, HuLuc63 exhibited antimyeloma activity against MM cells via NK cell-mediated ADCC in vitro and in vivo [29,80]. Elotuzumab (previously known as HuLuc63), a humanized anti-SLAMF7 monoclonal antibody, exhibits potent tumor-killing activity against MM cells and has been promoted to clinical development [81,82]. Elotuzumab recognizes an epitope within the membrane proximal IgC2 domain of SLAMF7 [83] and exhibits two main mechanisms of action. First, elotuzumab binds primarily to SLAMF7 on MM cells and subsequently kills these cells through Fc-gamma receptor (FcγR) III (CD16)-dependent ADCC mediated by NK cells. Second, elotuzumab directly activates NK cells to eliminate MM cells (Figure 4B,C) [81,83,84]. In the latter mechanism, the binding of elotuzumab to NK cells co-stimulates calcium signaling responses triggered through the NK receptors NKp46 and NKG2D in a CD16-independent manner [85]. Pazina et al. have recently reported that elotuzumab enhanced the SLAMF7–SLAMF7 interactions between NK cells and MM cells in a CD16-independent manner, which induced the cytotoxicity of the former toward the latter population (Figure 4D) [86]. Furthermore, Kurdi and colleagues reported a new mechanism wherein elotuzumab exhibited antimyeloma effects via macrophage-mediated antibody-dependent cellular phagocytosis (ADCP), which was triggered by the Fc- and CD16-dependent interaction of elotuzumab with macrophages (Figure 4E) [87]. However, elotuzumab does not trigger CDC against MM cells and cannot directly induce MM cell death [83]. Despite the promising effects of elotuzumab in preclinical studies, monotherapy with this immunotherapeutic agent failed to elicit objective responses in patients with relapsed/refractory MM due to NK-cell exhaustion in the tumor microenvironment [88].

In preclinical studies, lenalidomide or bortezomib was shown to potentiate the NK cell-mediated ADCC effect of HuLuc63 against MM cells [80,89]. The phase III ELOQUENT-2 study (NTC01239797) assessed the effects of elotuzumab when combined with lenalidomide and dexamethasone in 321 patients with relapsed/refractory MM, who were compared with 325 control patients treated with lenalidomide/dexamethasone alone. Treatment with the triple-combination regimen yielded a higher overall response rate (ORR; 79% vs. 66%) and very good partial response (VGPR) rate (33% vs. 28%) along with a consistently superior PFS benefit during a 4-year follow-up period [90,91]. The combination of elotuzumab with pomalidomide and low-dose dexamethasone in the ELOQUENT-3 trial (NCT02654132) yielded a markedly higher ORR rate (53% vs. 26%) and superior VGPR rate (20% vs. 9%) when compared with the control therapy of pomalidomide/low-dose dexamethasone [92]. Furthermore, combined elotuzumab, pomalidomide, and low-dose dexamethasone yielded a superior PFS outcome (median, 10.3 vs. 4.7 months with pomalidomide/dexamethasone) [92]. Both elotuzumab, lenalidomide, and dexamethasone and elotuzumab, pomalidomide, and dexamethasone combination regimens have been approved for the treatment of MM by the US Food and Drug Administration (FDA).

Although many reports have described NK-cell activation mediated via EAT-2-induced signals through the SLAMF7–SLAMF7 interaction, the adaptor protein-mediated signaling pathway downstream of SLAMF7 in MM cells remains unclear due to the deficiencies of both SAP and EAT-2 in these cells [69,78]. In EAT-2-deficient NK cells, SLAMF7 triggers tyrosine phosphorylation of SHIP-1 via CD45-dependent activation of Src kinases, inhibiting NK-cell function [93]. That is, the inhibitory signaling through SLAMF7 and SHIP-1 is required for the initiation of signaling via Src kinase activation mediated by CD45. Meanwhile, SLAMF7 cannot interact with SHIP-1 in MM cells that lack CD45, and thus the inhibitory signal mediated by SLAMF–7SHIP-1 is not induced in MM [93].

Recently, Kikuchi et al. have reported the homophilic binding of soluble SLAMF7 (sSLAMF7) to surface SLAMF7 molecules on MM cells and subsequently determined that this binding facilitated cell proliferation via SHP-2 and ERK signaling (Figure 4F) [94]. In this context, SLAMF7 knockdown suppressed MM cell growth by reducing the interactions between sSLAMF7 and SLAMF7 [94]. Those results suggest that SLAMF7 contributes to the pathogenesis of MM via SHP-2 and ERK signaling. Consistent with those results, we detected sSLAMF7 in the sera of 30% (32/103) of patients with newly diagnosed MM and observed significantly increased levels of this soluble protein in those with advanced R-ISS-stage disease [95]. Patients with detectable levels of sSLAMF7 exhibited aggressive clinical disease characteristics and had shorter PFS durations relative to their sSLAMF7-negative counterparts. The observation of decreased serum levels of sSLAMF7 in MM patients after elotuzumab treatment suggests that this drug might rapidly neutralize sSLAMF7 and suppress its ability to bind to SLAMF7 and induce signaling in MM. Kikuchi and colleagues demonstrated that treatment with lenalidomide and pomalidomide downregulated the expression of SLAMF7 on MM cells via the degradation of the transcription factor Ikaros. Moreover, the combination of elotuzumab and lenalidomide suppressed the ability of sSLAMF7 to promote MM cell proliferation in a xenograft mouse model [94]. Those findings suggest that elotuzumab, lenalidomide, and dexamethasone and elotuzumab, pomalidomide, and dexamethasone combination regimens could reasonably enhance the NK-mediated ADCC response while suppressing sSLAMF7-induced MM cell growth in vivo (Figure 4B).

Xie et al. determined the overexpression of SLAMF7 in MM cell lines harboring the t(4;14)(pl6.3;q32) translocation, which itself led to the overexpression of the multiple myeloma SET domain (MMSET) [96]. MMSET regulates the transcription of *SLAMF7*, and knockdown of the former was shown to downregulate the expression of the latter. The observation that SLAMF7 knockdown reduced colony formation in t(4;14) MM cell lines by inducing cell cycle G1 arrest and apoptosis suggests a potential association of SLAMF7 with the malignant progression of MM [96]. Post-hoc analyses of the ELOQUENT-2 study suggest that elotuzumab, lenalidomide, and dexamethasone therapy is beneficial for patients with relapsed/refractory t(4;14) MM [90,97].

Various CAR-T cell therapies for MM are currently under development [66]. For example, Chu and colleagues used primary human T cells and a retroviral vector to engineer SLAMF7-targeting CAR-T cells, which exhibited potent cytotoxic effects against both SLAMF7^high^ MM cell lines and primary MM cells and eliminated MM cells in MM.1S xenograft-bearing mice [98]. Gogishvili et al. developed SLAMF7 CAR-T cells engineered to express CAR constructs composed of a single-chain variable fragment (scFv) derived from huLuc63 [99]. Those CAR-T cells also recognized and killed SLAMF7^high^ primary MM cells from relapsed/refractory patients and led to prolonged survival in an MM xenograft mouse model. SLAMF7 CAR-T cells did not preclude SLAMF7^low/negative^ lymphocytes but induced selective autologous killing of SLAMF7^high/positive^ NK cell, T, and B cells [99]. Phase I clinical studies of SLAMF7 CAR-T therapy for relapsed/refractory MM patients are ongoing (NCT03958656 and NCT03710421). However, the translation of this therapeutic approach into real-world clinical settings may require gene editing to minimize the losses of SLAMF7-positive CAR-T cells and normal lymphocytes [99,100]. To overcome this problem and the limitation of the “per-patient basis” approach to autologous CAR-T therapy, a universal “off-the-shelf” allogeneic CAR-T, UCARTCS1, is under development [100]. UCARTCS1 is an allogeneic SLAMF7-targeting CAR-T in which TALEN genome editing technology was used to engineer a double-knockout of the genes encoding TCRα (*TRCA*) and SLAMF7 [101]. This approach prevents the development of graft-versus-host disease and reduces the risk of autologous killing of normal hematopoietic cells. UCARTCS1 exhibited striking and specific antimyeloma responses in a preclinical study, and a phase I clinical trial is ongoing (NCT04142619). Taken together, the evidence indicates that SLAMF7 is associated with the pathophysiology of MM and that therapies targeting this receptor could effectively treat relapsed/refractory MM.

## 7. Conclusions

Although recently FDA-approved novel treatments for MM have yielded improvements in both complete response rates and survival durations (from 3–4 to 7–10 years) [10], some patients experience repeated relapses of MM and may have refractory or incurable disease. Recent preclinical and clinical studies have revealed promising immunotherapies for MM. The SLAM family receptors SLAMF2, SLAMF3, SLAMF6, and SLAMF7 are strongly expressed on MM cells, and SLAMF3 and SLAMF7 play crucial roles in MM pathogenesis. Therefore, these receptors appear to be suitable potential immunotherapy targets in MM. For example, the monoclonal antibody elotuzumab promotes cytotoxicity against MM cells via NK-mediated ADCC but also directly inhibits MM cell growth by inhibiting sSLAMF7–SLAMF7 interactions. Elotuzumab may be particularly effective against t(4;14)-positive MM by targeting MMSET-mediated SLAMF7 overexpression. SLAMF3 is expressed at much higher levels on MM cells and progenitor cells (i.e., cancer stem cells) than on normal immune cells and HSCs/HPCs, and a preclinical study demonstrated the potential therapeutic efficacy of SLAMF3 CAR-T therapy. SLAMF3-targeting therapies may prevent further MM relapses by eliminating progenitor cells, leading to longer survival durations. Further studies are needed to clarify the expression and biological functions of SLAMF2 and SLAMF6 in MM cells and provide information to support the development of targeted immunotherapies (e.g., ADCs). The body of research outlined in this review will continue to support the development of various SLAM family receptor-targeting immunotherapies and ultimately improve the prognosis of patients with relapsed/refractory MM.

## Figures and Tables

**Figure 1 cancers-13-00279-f001:**
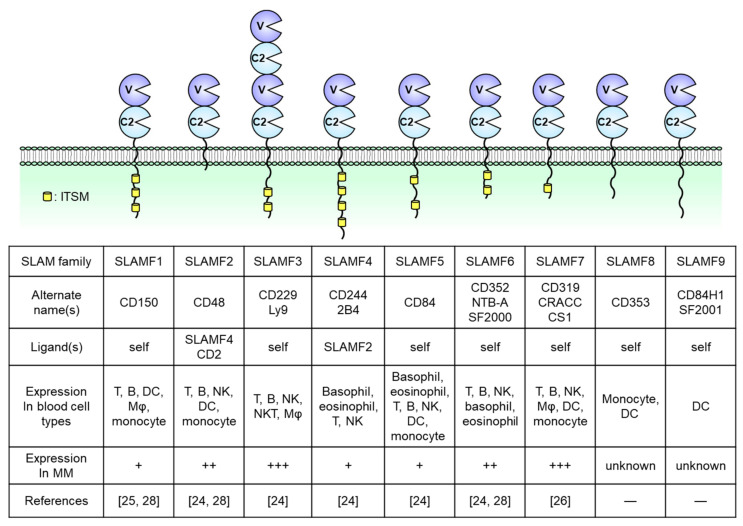
Structures of nine signaling lymphocytic activation molecule (SLAM) family receptors. The extracellular regions contain IgV (V)- and IgC2 (C2)-like domains. The receptors bind to self-ligands with high affinity through the IgV-like domain. In all family members except for SLAMF2, SLAMF8, and SLAMF9, the cytoplasmic tail contains an immunoreceptor tyrosine-based switch motif (ITSM). The expression of immune cells is from the database of the Human Protein Atlas (https://www.proteinatlas.org/). DC, dendritic cell, Mφ, macrophage, NK, natural killer.

**Figure 2 cancers-13-00279-f002:**
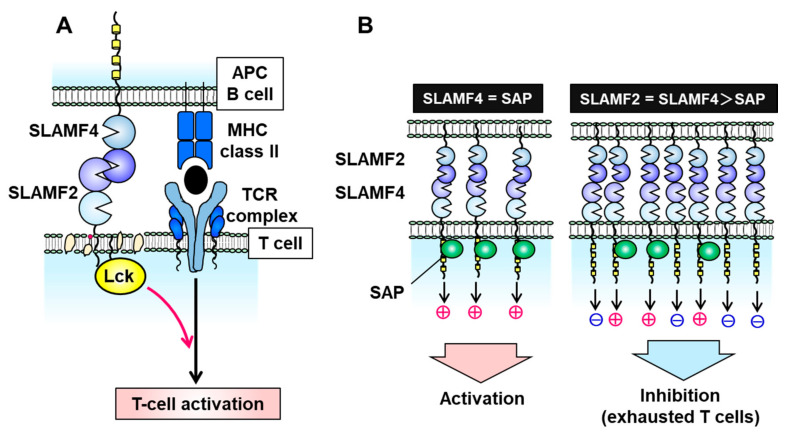
Signal transduction via the signaling lymphocytic activation molecule family receptor (SLAMF) 2–SLAMF4 interaction. (**A**) On T cells, SLAMF2 promotes T cell receptor (TCR) signaling and T cell activation by binding to SLAMF4 on antigen-presenting cells (APCs) and activating the tyrosine kinase Lck in T cell membrane lipid rafts. (**B**) The SLAMF2–SLAMF4 interaction facilitates natural killer (NK) or T cell activation in the presence of sufficient levels of the adaptor protein SAP but induces an inhibitory signal in the absence of SAP.

**Figure 3 cancers-13-00279-f003:**
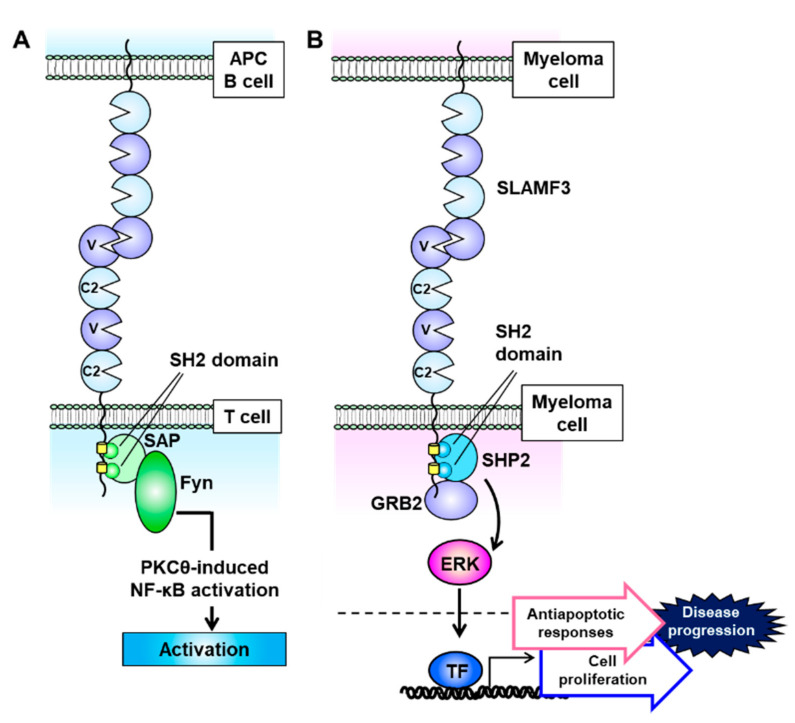
Signaling lymphocytic activation molecule family receptor (SLAMF) 3 signal transduction is mediated via self-ligation and cell–cell interactions. (**A**) SLAMF3 recruits SAP at two immunoreceptor tyrosine-based switch motif (ITSM) domains, and the SLAM–SAP–Fyn signaling pathway promotes T cell activation. (**B**) On multiple myeloma (MM) cells, SLAMF3 activates the ERK signal transduction pathway. This function is mediated via self-ligand interactions between MM cells and SHP2 and GRB2 formation, which accelerates cell proliferation and antiapoptotic signaling to increase the malignant potential of MM.

**Figure 4 cancers-13-00279-f004:**
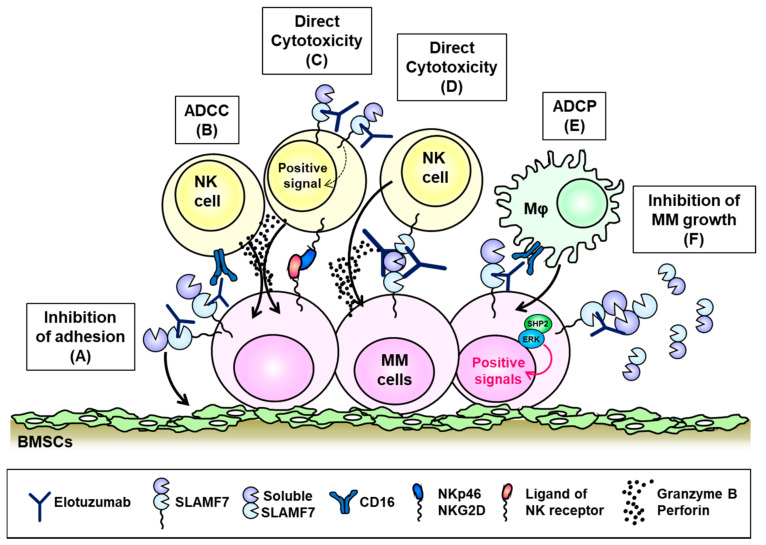
The biological function of the signaling lymphocytic activation molecule family receptor (SLAMF) 7 on multiple myeloma (MM) cells and mechanism of action of elotuzumab. (**A**) On MM cells, SLAMF7 is associated with adhesion to bone marrow stroma cells (BMSCs) and elotuzumab blocks MM cell adhesion to BMSCs. Elotuzumab stimulates robust natural killer (NK) cell-mediated antibody-dependent cell-mediated cytotoxicity (ADCC) via the Fc–CD16 interaction (**B**), directly induces activating signal transduction in NK cells (**C**), improves NK-cell-mediated cytotoxicity in response to the SLAMF7–SLAMF7 interaction (**D**), and enhances macrophage (Mφ)-mediated antibody-dependent cellular phagocytosis through the CD16 receptor (**E**). (**F**) SLAMF7 promotes MM cell growth via SHP2–ERK signaling in response to binding with soluble SLAMF7 (sSLAMF7), and elotuzumab suppressed MM cell growth promoted by SLAMF7–sSLAMF7 interactions in vitro.

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
