# Peer review of "Immune Functions of Signaling Lymphocytic Activation Molecule Family Molecules in Multiple Myeloma"

_cancers, 2021, doi:10.3390/cancers13020279_

Round 1

Reviewer 1 Report

Several reviews are available on the role of signaling lymphocytic activation molecule (SLAM) family in multiple myeloma (MM). However, this manuscript is complete and details the expression and function of the SLAM family receptors during the malignant progression of MM. Furtheremore, preclinical and clinical studies that could candidate these receptors as potential immunotherapeutic targets in MM were reported in this review. The manuscript is well designed and together with the figures, the authors explain the biological role of SLAM family receptors in the context of MM .

Please consider revising the manuscript to correct the language and style to make it more grammatically accurate.

Author Response

To Reviewer #1

Several reviews are available on the role of signaling lymphocytic activation molecule (SLAM) family in multiple myeloma (MM). However, this manuscript is complete and details the expression and function of the SLAM family receptors during the malignant progression of MM. Furthermore, preclinical and clinical studies that could candidate these receptors as potential immunotherapeutic targets in MM were reported in this review. The manuscript is well designed and together with the figures, the authors explain the biological role of SLAM family receptors in the context of MM .

Comment:

Please consider revising the manuscript to correct the language and style to make it more grammatically accurate.

Response:

Thank you for the suggestions. The grammatical and stylistic problems were corrected in the revised version by specialized language-editing service.

Reviewer 2 Report

The authors present a comprehensive review on expression and biological functions of signaling lymphocytic activation molecule family receptors and their significance in the treatment of multiple myeloma. The paper particularly focuses on SLAMF2, SLAMF3, SLAMF6, and SLAMF7.Overall, the manuscript is well written and covers all aspects of SLAM as potential target in the treatment of multiple myeloma.

Some comments:

1) Page 2, line 45: CD38 is targeted not only by daratumumab but also by isatuximab.

2) Page 2, paragraph 2: The four characteristics of potential target antigens listed are general attributes that should be given for a successful immunotherapy. However, the four properties appear hardly discovered by the authors (we have identified). Please adjust.

3) Page 9, line 335-7: What do the authors mean by stating “..yielded few additional clinical benefits..” As far as I am aware, the combination of elotuzumab, bortezomib and dexamethasone was not further pursued primarily for business policy decisions, and not for reasons of insufficient efficacy.

Author Response

To Reviewer #2

The authors present a comprehensive review on expression and biological functions of signaling lymphocytic activation molecule family receptors and their significance in the treatment of multiple myeloma. The paper particularly focuses on SLAMF2, SLAMF3, SLAMF6, and SLAMF7.Overall, the manuscript is well written and covers all aspects of SLAM as potential target in the treatment of multiple myeloma.

Comment 1:

Page 2, line 45: CD38 is targeted not only by daratumumab but also by isatuximab.

Response:

According to the reviewer’s comment, we added isatuximab in the sentence on page 2, lines 48 and 54.

Comment 2:

Page 2, paragraph 2: The four characteristics of potential target antigens listed are general attributes that should be given for a successful immunotherapy. However, the four properties appear hardly discovered by the authors (we have identified). Please adjust.

Response:

As pointed out by the reviewer, the text in paragraph 2, page 2, was not appropriate and we revised this text (page 2, lines 5766).

Comment 3:

Page 9, line 335-7: What do the authors mean by stating “..yielded few additional clinical benefits..” As far as I am aware, the combination of elotuzumab, bortezomib and dexamethasone was not further pursued primarily for business policy decisions, and not for reasons of insufficient efficacy.

Response:

We thank the reviewer for pointing this out and deleted this sentence.

Reviewer 3 Report

The authors provide a very nice review of SLAM molecules in MM, with some well made figures.  The review is comprehensive with particular focus on SLAMF7 which is a key therapeutic target in MM.  I don't spot any problems with the content or language.

Author Response

To Reviewer #3

The authors provide a very nice review of SLAM molecules in MM, with some well made figures. The review is comprehensive with particular focus on SLAMF7 which is a key therapeutic target in MM.  I don't spot any problems with the content or language.

Response:

We thank the reviewer for this comment.

Reviewer 4 Report

This is a solid and useful review of the expression of several members of the SLAM family of receptors in multiple myeloma, their involvement in disease, and strategies being used to therapeutically target these receptors to treat patients. It is very well written, although a few minor suggestions and correction of a couple misconceptions would improve the manuscript.

  1. It would be helpful to have a bit more background info in the introduction (probably around line 78) on signaling mechanisms downstream from EAT-2 and SAP, plus specifics on immune cell type expression of each of these, and which bind to each of the SLAM receptors discussed (perhaps in Figure 1), since these differ. This only needs to be a paragraph, but would provide important background information to delineate differential signaling through different SLAMs in different immune cell types. 
  2. Reference 46 is misquoted at line 134. They did not use a syngeneic (written “synergetic”) mouse model for the functional studies, but SCID or NOD/SCID mice with human myeloma cell lines in that study. 
  3. Also, there are misconceptions on EAT-2 and CD45 involvement in SLAMF7 signaling. Line 338 suggests that NK cells lack both SAP and EAT-2, but in fact NK cells are one of the few immune cell types that constitutively express EAT-2, and EAT-2 is the required signaling adaptor for SLAMF7 signaling (ref. #75).  Reference #92 used EAT-2 deficient NK cells to show that SLAMF7 acts as an inhibitory receptor in the absence of EAT-2, but this was using NK cells from genetically-modified EAT-2 knockout mice.  Also, an important point from this paper is that myeloma cells lack expression of EAT-2 and SAP, thus SLAMF7 would be expected to exhibit inhibitory function via recruitment of SHIP-1 in this genetically modified context.  Myeloma cells, however, lack CD45, which is a phosphatase that is required to achieve an active form of src family kinases, which phosphorylate the tyrosine binding sites on SLAMF7 to recruit SHIP-1.  Thus, in myeloma cells, SHIP-1 signaling through SLAMF7 is inactivated, since the tyrosines cannot be phosphorylated, at least efficiently (although SHP-2-ERK signaling is apparently still intact).  These important points need to be included.
  4. A couple minor typos were noted:
    1. Line 341: this inhibitor signal
    2. Line 361: harboring the t(4;14)…..

Author Response

To Revewer #4

This is a solid and useful review of the expression of several members of the SLAM family of receptors in multiple myeloma, their involvement in disease, and strategies being used to therapeutically target these receptors to treat patients. It is very well written, although a few minor suggestions and correction of a couple misconceptions would improve the manuscript.

Comment 1:

It would be helpful to have a bit more background info in the introduction (probably around line 78) on signaling mechanisms downstream from EAT-2 and SAP, plus specifics on immune cell type expression of each of these, and which bind to each of the SLAM receptors discussed (perhaps in Figure 1), since these differ. This only needs to be a paragraph, but would provide important background information to delineate differential signaling through different SLAMs in different immune cell types. 

Response:

According to the reviewer’s comment, we added text on the expression and signaling pathway of the adaptor protein SAP and EAT-2 in Section 2 (pages 2–3, lines 84–97).

Comment 2:

Reference 46 is misquoted at line 134. They did not use a syngeneic (written “synergetic”) mouse model for the functional studies, but SCID or NOD/SCID mice with human myeloma cell lines in that study.

Response:

We thank the reviewer for pointing out this mistake and revised this sentence (page 5, lines 166–168).

Comment 3:

Also, there are misconceptions on EAT-2 and CD45 involvement in SLAMF7 signaling. Line 338 suggests t misconceptions hat NK cells lack both SAP and EAT-2, but in fact NK cells are one of the few immune cell types that constitutively express EAT-2, and EAT-2 is the required signaling adaptor for SLAMF7 signaling (ref. #75).  Reference #92 used EAT-2 deficient NK cells to show that SLAMF7 acts as an inhibitory receptor in the absence of EAT-2, but this was using NK cells from genetically-modified EAT-2 knockout mice.  Also, an important point from this paper is that myeloma cells lack expression of EAT-2 and SAP, thus SLAMF7 would be expected to exhibit inhibitory function via recruitment of SHIP-1 in this genetically modified context.  Myeloma cells, however, lack CD45, which is a phosphatase that is required to achieve an active form of src family kinases, which phosphorylate the tyrosine binding sites on SLAMF7 to recruit SHIP-1.  Thus, in myeloma cells, SHIP-1 signaling through SLAMF7 is inactivated, since the tyrosines cannot be phosphorylated, at least efficiently (although SHP-2-ERK signaling is apparently still intact).  These important points need to be included.

Response:

We are sorry to have caused a misunderstanding by stating “NK cells lack both SAP and EAT-2 expression” and revised this sentence in section 6 (page 10, lines 383–386). According to the reviewer’s suggestions, we added text on the inhibitory signaling of SLAMF7–SHIP-1 in NK and MM cells (page 10, lines 386–392).

Comment 4:

A couple minor typos were noted:

  1. Line 341: this inhibitor signal

Response: We deleted this sentence in our manuscript.

  1. Line 361: harboring the t(4;14)…..

Response:

We corrected this as you pointed out (page 10, lines 414–415).
